# Study on Seismic Risk Assessment Model of Water Supply System in Chinese Mainland

Tianyang Yu[1,2]   Banghua Lu[1,2]   Hui Jiang[1,3]   Zhi Liu[1,3]

1. Guangdong Earthquake Agency; Guangzhou, China
2. Guangdong Earthquake Disaster Risk Control and Prevention Center, Guangzhou, China
3. Shenzhen Academy of Disaster Prevention and Reduction; Shenzhen, China

Correspondence: Tianyang Yu (821677781@qq.com), Banghua Lu (76415990@qq.com) and Zhi Liu (liuzhi8725@126.com).

**Abstract:** Using the PGA under four exceeding probabilities calculated by China probabilistic seismic hazard analysis method, the probability density function of PGA was obtained by fitting utilizing the Cornell seismic hazard exceeding probability-PGA function model. Combined with the seismic fragility function of the water supply system calculated based on the empirical matrix of actual earthquake damage and the exposure of fixed assets, the seismic loss expectation and loss rate expectation models of the water supply system were established, and the classification standard with the seismic loss rate expectation of the water supply system as the index was proposed. The seismic fragility of the water supply system was classified, and the exposure of the water supply system was analyzed. The total fixed assets in the Water Supply Yearbook were taken as the exposure to earthquake in the region. The accuracy of the fragility model in this paper was verified through the actual earthquake damage losses in Deyang City. Taking the water supply system of 720 cities in Chinese Mainland as an example, the distribution maps of seismic loss expectation and loss rate expectation were calculated and drawn. The loss rate expectation model was verified by the key earthquake prevention areas in Chinese Mainland. The assessment model based on loss expectation and loss rate expectation was taken as the seismic risk assessment model of water supply system in Chinese Mainland.

## Introduction

Today, with the gradual improvement of human civilization and material wealth, the increasing number of earthquake disaster around the world poses a huge threat to urban water supply systems. 40% of major cities in China are located near major earthquake zones, with 17% facing high risk, and 55% of cities may suffer serious disasters (Gao Mengtan, 2020). After a strong earthquake, as an important component of civil engineering, the urban water supply system and emergency rescue system in lifeline engineering are called lifelines in lifeline engineering. Therefore, to ensure the normal operation of lifelines after an earthquake, the government should increase investment and management (Nigg J, 1998). Once the water supply system is damaged by an earthquake, it not only cannot meet the normal water supply for residents, but also cannot provide water for emergency rescue departments and prevent the spread of fires. At the same time, the inability of enterprises to use

production water can also bring indirect economic losses. In 1994, the North Ridge 6.6 magnitude earthquake in the United States caused widespread rupture of over 1400 Los Angeles water supply pipelines, of which 100 were located on the main water supply network (Han Yang, 2002). The 1995 Kobe 7.3 magnitude earthquake in Japan caused damage to 1610 destruction of the main water supply system in the earthquake area, causing 80% of users in 9 cities water-break, 90% of water supply facilities in the Kobe area of Osaka to be damaged, and 120000 underground water supply pipelines to leak. At the same time, the interruption of water supply also seriously hindered firefighting work (He Weihua, 2009); The power failure of the Fukushima nuclear power plant caused by the March 11 earthquake in Japan led to the failure of the water supply system, which led to nuclear reactor meltdown. The 1976 Tangshan earthquake resulted in the paralysis of the city's water supply system, with a pipeline damage rate of 4 per kilometer. 332 main networks in Tanggu District were damaged, and after half a month of emergency repair, only 50% of the water supply capacity was restored (Han yang, 2002). The water supply system of Mianzhu City suffered devastating damage in the 2008 Wenchuan 8.0 earthquake (Institute of Engineering Mechanics, CEA, 2009). Research has shown that the indirect economic losses caused by water supply interruptions are often dozens of times greater than the direct economic losses caused by earthquake damage in the water supply system (Brozovic N, 2007). Therefore, the importance and urgency of building a regional and urban water supply system seismic risk assessment model to provide decision-making basis for the government and business departments has emerged.

In the 1984 UNESCO research plan (Jiang Hui, etc.,2022), Varnes proposed a definition of natural disasters and risks, which has been widely recognized by experts in the field of natural disaster research both domestically and internationally. The basic model of earthquake (disaster) risk assessment also conforms to this definition. At present, scholars at home and abroad have different definitions of the concept of earthquake disaster risk. The commonly used earthquake disaster risk refers to the possibility of damage and loss to buildings (structures) or lifeline projects in specific areas in the future within a certain time limit, as well as the possibility of loss to life, property, national economy, etc., which can be expressed as:

$$R = f\left(H, E, V\right)$$

Taking into account the impact of site conditions, the above equation can be further expressed as:

$$R = H \cdot E \cdot V \cdot S$$

In the formula, R is the risk of earthquake (disaster), referring to the potential losses caused by future earthquakes; H refers to the seismic hazard, which refers to the probability of future earthquakes occurring within a certain region within a certain period of time; E is the value of the disaster bearing body or social wealth, which refers to the exposure level of the disaster bearing body (including buildings, lifeline engineering, population, property, etc.) threatened by earthquakes in a given area; V is the vulnerability of the disaster bearing body under earthquake action, or the loss rate of the disaster bearing body under different earthquake intensities, which can be

represented by a number between 0 and 1 (0 represents no loss, 1 represents complete loss); S is the site impact coefficient.

The risk assessment research in this paper was based on the above three elements of earthquake disaster risk (seismic hazard, vulnerability of disaster bearing body, and asset exposure) to establish a risk assessment model based on the loss rate expectation of water supply system. Based on this approach, we carry out data collection, organization, modeling, and other work. The flow chart of seismic risk assessment for water supply systems can be seen in Figure 1.

**Figure 1 Flow Chart of Seismic Risk Assessment for Water Supply Systems**

## 1. Basic database for risk assessment

The risk assessment data involved in this study includes regional basic data of the water supply system, including five categories. The first category is the material of the water supply pipeline network extracted from the "Water Supply Yearbook"(Statistical Yearbook of Urban Water Supply (2009-2018)). The second category is the urban basic fortification intensity extracted from the "Seismic Code"(GB50011-2010 Code for seismic design of buildings. (2010).). The third

category is the urban population, GDP and other data extracted from the Census (National Bureau of Statistics of China. (2011).), which have been processed to provide urban classification. The fourth category is site classification. The fifth category is seismic hazard data extracted from the "Fifth Zonation Map"(GB18306-2015Seismic ground motion parameters zonation map of China. (2015).). The above basic data covers 720 cities in 31 provinces and autonomous regions except Taiwan, Hong Kong, and Macau.

(1) Water supply system

This paper is mainly based on the pipeline material data in 2018 Water Supply Yearbook, and mainly collects the length data of five pipeline materials, namely, Ductile Cast iron pipe, steel pipe, Cast iron pipe, prestressed reinforced concrete pipe and plastic pipe. At present, data from a total of 720 cities has been compiled. Although the data covers 31 provinces and cities in mainland China, there are differences in data coverage for each province. The western region does not have complete data for the eastern region, such as Qinghai and Tibet, which only have data for one city each.

(2) Fortification intensity data

This article extracts the seismic fortification intensities of 720 cities that have been organized in the "Seismic Code".

(3) City category data (population and GDP)

Extract urban category data based on the urban population and GDP data from the 6th National Population Census released by the national statistical department. Determine the city categories of 720 cities through certain data processing methods.

(4) Site Category Data

In the national site classification database established using the BP neural network site classification method (Allen, T. I., and Wald, D. J. (2007). Shi, D. C. (2009). Yu Haiying and Ma Wenxi.(2020).), 720 site categories representing the city's water supply system were extracted.

(5) Seismic hazard data

According to the determined potential source area division scheme, seismicity parameter scheme and ground motion parameter attenuation relationship, the peak

acceleration $a_{Ei}$ under four different exceeding probability levels of basic ground

motion, frequent ground motion, rare ground motion and extremely rare ground motion in I1 site category of grid averaged distribution sites nationwide was given by using the probabilistic seismic hazard analysis method and the basic database of the

Fifth Generation Zonation Map. The grid density is $0.1° \times 0.1°$. This article extracted

seismic hazard data for government residences in 720 cities from the database. Taking Heyuan city as an example, seismic hazard raw data could be seen in Table 1. The probability density function of the PGA of 720 cities was calculated by the piecewise fitting method of the seismic hazard curve.

**Table 1 Seismic Hazard Data of Heyuan City (Raw Data of 4 Probability**

| Control Points) | | | | |
|---|---|---|---|---|
| 50 year exceeding probability | 63% | 10% | 2% | 0.5% |
| PGA(gal) | 19.6 | 71.6 | 172.4 | 296.6 |

Among the 5 types of data in the above databases, the water supply networks data from the Water Supply Yearbook, the seismic fortification intensity of the Seismic Code, the population and GDP data from the Census do not require complex processing for this study. However, the site category data needs to be analyzed for accuracy and usability, and the seismic hazard data needs to be processed using seismic hazard analysis methods for this study. Taking the basic data of Heyuan City as an example, the database structure is shown in Table 1 and Table 2.

**Table 2 Basic Data of Water Supply Network in Heyuan City**

| City code | Province Code | City | Province | Longitude | Latitude | Site category | City category | Fortification intensity |
|---|---|---|---|---|---|---|---|---|
| 441600 | 440000 | Heyuan | Guangdong | 114.692 | 23.7367 | II | 3 | 7 |

**Table 2(continuous) Basic Data of Water Supply Network in Heyuan City**

| Pipe category | Ductile cast iron pipe | Steel pipe | Plastic pipe | Prestressed reinforced concrete pipe | Cast iron pipe |
|---|---|---|---|---|---|
| Pipe length of water supply network (kilometers) | 48.96 | 84.23 | 289.16 | 41.3 | 15 |

This article collected seismic damage data from cities such as Haicheng, Tangshan, and Wenchuan (Institute of Engineering Mechanics, CEA.,1979. Institute of Engineering Mechanics, CEA.,2009.) and classified, organized and calculated the seismic damage matrices of water supply pipelines, water tanks, and pump houses according to the city classification and seismic damage data. A database of seismic damage data for water supply systems was established.

After sorting, the seismic damage rates of different materials of water supply pipelines in the Haicheng earthquake are shown in Table 3. The water supply pipeline materials are mainly cast iron pipes.

**Table 3 Seismic damage rates of different pipeline materials in Haicheng earthquake (location/10 kilometers)**

| City | Steel pipe | Asbestos cement pipe | Cast iron pipe |
|---|---|---|---|
| Panshan(VII) | 70.0 | 13.0 | 16.0 |
| Yingkou city(VIII) | 114.0 | 20.0 | 10.6 |
| Yingkou town(IX) | 21.0 | 70.0 | 12.3 |

| Haicheng(IX) | 157.0 | 90.0 | 212.0 |
|---|---|---|---|

The seismic damage rates of the water supply pipelines during the Tangshan earthquake was summarized in Table 4. Water supply pipelines include cast iron pipes, prestressed reinforced concrete pipes, steel pipes, and self stressing reinforced concrete pipes, with cast iron pipes accounting for the largest proportion.

**Table 4 Seismic damage rates of water supply network in Tangshan earthquake (location/kilometer)**

| City | Pipe length(km) | Diameters(mm) | Average damage rate (location/km) |
|---|---|---|---|
| Tianjin(VII~VIII) | 870 | 75~1000 | 0.18 |
| Tanggu(VIII) | 79.5 | 75~600 | 4.18 |
| Hangu(IX) | - | - | 10 |
| Tangshan(IX~X) | 111 | 75~600 | 4 |

After sorting, the seismic damage rates of various pipes in the water supply network during the Wenchuan earthquake are shown in Table 5.

**Table 5 Seismic damage rates of water supply pipelines during the Wenchuan earthquake (location/10km)**

| Seismic intensity | Steel pipe | Cast iron pipe | Cement pipe | PE pipe | Ductile cast iron pipe | PVC pipe |
|---|---|---|---|---|---|---|
| VI | 0 | 1.50 | 0 | 0 | 0 | 0 |
| VII | 0.60 | 12.90 | 8.30 | 3.00 | 0.34 | 6.14 |
| VIII | 22.30 | 40.00 | 20.36 | 8.00 | 1.20 | 25.00 |

2)Water reservoir(Clean water reservoir and water treatment reservoir)

We have compiled seismic damage data for 200 clean water reservoirs and 124 water treatment reservoirs in the Haicheng earthquake, Tangshan earthquake, Baotou West earthquake, Yutian-Cele earthquake in Xinjiang, Wenchuan earthquake, and Yushu earthquake (Gao Lin, 2012). The seismic damage statistics are shown in Tables 6 and 8; The seismic damage matrix of the clean water reservoir and water treatment reservoir is shown in Tables 7 and 9.

**Table 6 Statistical table of seismic damage of clean water reservoir**

| Damage level | Basically intact | Slight damage | Moderate damage | Severe damage | Destroyed |
|---|---|---|---|---|---|
| Total(seats) | 156 | 15 | 12 | 14 | 3 |

**Table 7 Seismic damage matrix of clean water reservoir(%)**

| Seismic intensity | Basically intact | Slight damage | Moderate damage | Severe damage | Destroyed |
|---|---|---|---|---|---|
| VI | 85 | 15 | 0 | 0 | 0 |
| VII | 76 | 19 | 5 | 0 | 0 |

| | | | | | |
|---|---|---|---|---|---|
| VIII | 19 | 29 | 33 | 15 | 4 |
| IX | 8 | 12 | 43 | 28 | 9 |
| X | 0 | 0 | 25 | 45 | 30 |

**Table 8 Statistical table of seismic damage of water treatment reservoir**

| Damage level | Basically intact | Slight damage | Moderate damage | Severe damage | Destroyed |
|---|---|---|---|---|---|
| Total(seats) | 97 | 8 | 10 | 8 | 1 |

**Table 9 Seismic damage matrix of water treatment reservoir(%)**

| Seismic intensity | Basically intact | Slight damage | Moderate damage | Severe damage | Destroyed |
|---|---|---|---|---|---|
| VI | 92 | 7 | 1 | 0 | 0 |
| VII | 64 | 21 | 12 | 3 | 0 |
| VIII | 33 | 26 | 22 | 13 | 6 |
| IX | 0 | 0 | 35 | 45 | 20 |

3) Pump station building

This article uses the seismic damage matrix of pump station buildings obtained through actual seismic damage statistical analysis as the basic seismic damage data for the fragility curves. The seismic damage matrix of pump buildings can be found in the literature "Research on New Techniques for Evaluating the Loss of Large Earthquake Disasters in Water Supply Systems" (Institute of Engineering Mechanics, China Earthquake Administration, 2013).

The above basic data constitute the basic database for seismic risk assessment of water supply system.

## 2 Seismic risk assessment model based on loss (rate) expectation

The seismic loss expectation is expressed by the coupling of three factors: seismic hazard, structural vulnerability and social wealth (Chen Yong, 1999): as an expression of earthquake disaster risk, the seismic risk loss (rate) expectation refers to the intersection of seismic hazard, structural vulnerability of water supply system facilities and total fixed assets of water supply system in a certain region in a certain period of time in the future.

2.1 Seismic Hazard

The process of seismic hazard probability analysis includes complex earthquake repetition models and earthquake motion prediction models, but the expression of seismic hazard analysis results is not complex and is generally represented by seismic hazard curves. The seismic hazard curve should provide exceeding probability curve for the ground motion parameters, which is the probability of exceeding the given ground motion parameter value on the probability distribution curve. The seismic hazard curve is determined by the potential source and the attenuation law of ground motion parameters. In this paper, the probability density function of peak ground

acceleration was calculated by using the piecewise fitting method of seismic hazard
curve.

The relationship between the seismic hazard function $H_t(a)$ of the engineering

site and the peak ground acceleration $a$ is (Cornell, 1968):

$$H_t(a) = 1 - \exp(k_b t a^{k_H}) \qquad (1)$$

Where $a$ is peak ground acceleration, $t$ is Time (year), $k_b$ and $k_H$ is

Parameters of seismic hazard curve.
This article used certain designated control points piecewise fitting the seismic
hazard curve, while the exceeding probability of other PGA parameters was obtained
from the seismic hazard curve.
The probability seismic hazard analysis method compiled by the "Fifth Zonation
Map" was used to calculate the annual exceeding probability of the peak ground
acceleration of the rock site in Mengzi City, Yunnan Province (Wen Manhua, 2017),
as shown in Table 10.
**Table 10 PGA of rock at a certain site in Mengzi City-annual exceeding probabilities**

| PGA/gal | Annual exceeding probability | PGA/gal | Annual exceeding probability | PGA/gal | Annual exceeding probability |
|---|---|---|---|---|---|
| 1 | 4.12E-01 | 60 | 6.87E-03 | 200 | 1.46E-04 |
| 5 | 3.27E-01 | 70 | 4.68E-03 | 250 | 5.31E-05 |
| 10 | 1.58E-01 | 80 | 3.29E-03 | 300 | 2.02E-05 |
| 15 | 9.31E-02 | 90 | 2.38E-03 | 350 | 7.85E-06 |
| 20 | 6.02E-02 | 100 | 1.74E-03 | 400 | 2.98E-06 |
| 30 | 2.99E-02 | 125 | 8.64E-04 | 450 | 1.11E-06 |
| 40 | 1.70E-02 | 150 | 4.60E-04 | 500 | 3.95E-07 |
| 50 | 1.05E-02 | 175 | 2.55E-04 | 600 | 2.33E-08 |

18      The corresponding relationship between the peak ground acceleration of four
19  control points of a rock site in Mengzi City and the exceeding probability in different
20  time scales is shown in Table 11.
21      **Table 11 PGA of rock at a certain site in Mengzi City-exceeding probabilities**

| PGA/g | 37.92 | 94.31 | 156.80 | 224.76 |
|---|---|---|---|---|
| 1 year exceeding probability | 1.97% | 0.21% | 0.04% | 0.01% |
| 10 year exceeding probability | 18.03% | 2.08% | 0.40% | 0.10% |
| 50 year exceeding probability | 63.00% | 10.00% | 2.00% | 0.50% |
| 100 year exceeding probability | 86.31% | 19.00% | 3.96% | 1.00% |

23      According to Table 11, the parameters of the 1-year segmented seismic hazard
24  function for the rock site in Mengzi City were fitted using the least squares method, as
25  shown in Table 12. The data in Table 10 and the fitted 1-year seismic hazard curve for
26  the rock site were plotted in the same coordinate system, as shown in Figure 2.

**Table 12 Parameters of Seismic Hazard Function in Mengzi City**

| City | Fortification intensity | Site classification | Segmentation | 1-year | |
|------|------------------------|---------------------|--------------|--------|--------|
| | | | | $k_H$ | $k_b$ |
| | | | 1st segment | -2.47 | -156.2 |
| Mengzi | VII | $I_1$ | 2nd segment | -3.26 | -5841.5 |
| | | | 3rd segment | -3.85 | -113627.1 |

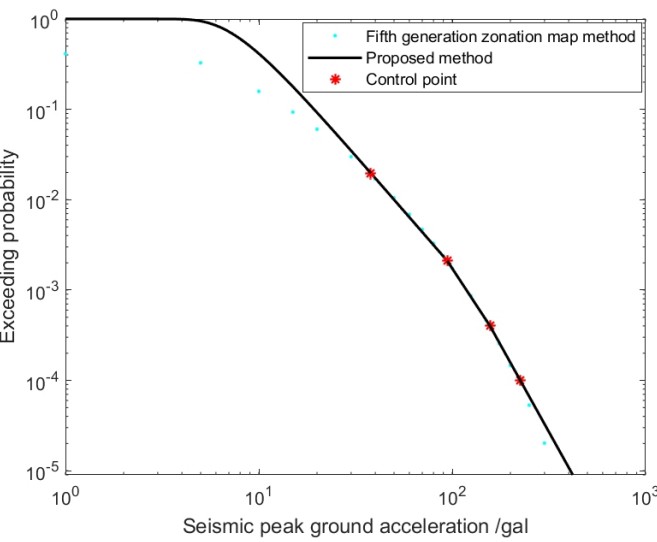

**Figure 2 Seismic hazard curve of 1-year rock site in Mengzi City**

From Figure 2, it can be seen that the seismic hazard curve obtained by the piecewise fitting method is basically consistent with the seismic hazard calculation points obtained by the fifth generation seismic zonation map method. When the peak ground acceleration is small, the annual exceeding probability will be overestimated. In fact, when the peak ground acceleration is small, the water supply system is basically in good condition, and its loss ratio is 0. Even if the exceeding probability is overestimated, the accuracy of the seismic risk analysis results of the water supply system will not be affected. Therefore, it is feasible to obtain seismic risk curve parameters in different regions of Chinese Mainland by piecewise fitting four control points given in the Fifth Generation Zonation Map (GB18306-2015 Seismic ground motion parameters zonation map of China. (2015).).

The ratio relationship between the PGA corresponding to the 50 year exceeding probability of 63%, the 50 year exceeding probability of 2%, and the annual exceeding probability of 10-4 and the basic ground motion PGA (50 year exceeding probability of 10%) is very complex, and its spatial distribution has a great correlation with the distribution of potential source areas, which is mainly affected by the seismotectonics environment, and the ratios in different regions vary greatly (Gao Mengtan, 2006; Lei Jiancheng, etc., 2010). Therefore, it is not possible to directly use the PGA(0.05g, 0.1g, 0.15g, 0.2g, 0.3g) corresponding to the 50 year exceeding probability of 10% of specific sites in the "Fifth Generation Zonation Map" to calculate the PGA under the other three exceeding probabilities in a fixed proportion. Instead, based on the basic database of the "Fifth Generation Zonation Map", further analysis and processing are conducted on the actual calculated seismic hazard data

(using CPSHA method) extracted from the database.

Since the PGA provided in the Fifth Generation Zonation Map is under a specific site category, it is necessary to obtain the PGA under the corresponding site category by interpolation and transformation according to the actual site category of the city using the method provided in the Fifth Generation Zonation Map. This paper collected seismic hazard data (four control points) and actual site categories of 720 cities in Chinese Mainland.

By using the relationship between the cumulative distribution function (CDF) and the exceeding probability, the functional relationship between the cumulative distribution function $C_t(a)$ and the PGA $a$ can be obtained as follows:

$$C_t(a) = 1 - H_t(a) = \exp\left(k_b t a^{k_H}\right) \quad\quad (2)$$

The probability density function (PDF) of PGA can be obtained by calculating the first derivative of the cumulative distribution function, that is, the functional relationship between $f_t(a)$ and the PGA $a$ is:

$$f_t(a) = \exp\left(k_b t a^{k_H}\right) \cdot k_b \cdot t \cdot k_H \cdot a^{k_H - 1} \quad\quad (3)$$

Based on the above method, the relevant parameters of the probability density function $f_t(a)$ of the PGA of 720 cities in 10-year, 50-year and 100-year scales under the actual site categories are calculated, and a seismic hazard database that can be used for the seismic risk assessment model is formed. This article listed the parameters of segmented seismic hazard functions at the 10 year, 50 year, and 100 year scales for the actual site categories of three typical cities, as shown in Table 13. The seismic hazard curves of four typical cities are plotted, as shown in Figures 3 to 6.

**Table 13 Parameters of Seismic Hazard Function for Example Cities**

| City | Site | Segmentation | 10-year | | 50-year | | 100-year | |
|---|---|---|---|---|---|---|---|---|
| | | | $k_H$ | $k_b$ | $k_H$ | $k_b$ | $k_H$ | $k_b$ |
| Heyuan | | 1st segment | -1.76 | -3.00E-05 | -1.76 | -3.00E-05 | -1.76 | -3.00E-05 |
| | II | 2nd segment | -1.86 | -2.37E-05 | -1.85 | -2.40E-05 | -1.85 | -2.40E-05 |
| | | 3rd segment | -3.76 | -1.31E-06 | -3.78 | -1.29E-06 | -3.77 | -1.30E-06 |
| Deyang | | 1st segment | -2.22 | -3.46E-05 | -2.21 | -3.40E-05 | -2.21 | -3.50E-05 |
| | III | 2nd segment | -4.08 | -1.08E-06 | -4.07 | -1.11E-06 | -4.07 | -1.11E-06 |
| | | 3rd segment | -4.92 | -3.26E-07 | -4.94 | -3.18E-07 | -4.93 | -3.22E-07 |
| Kelamayi | | 1st segment | -1.98 | -1.76E-05 | -1.98 | -1.77E-05 | -1.98 | -1.80E-05 |
| | II | 2nd segment | -2.49 | -5.17E-06 | -2.48 | -5.29E-06 | -2.48 | -5.29E-06 |
| | | 3rd segment | -3.13 | -1.68E-06 | -3.15 | -1.65E-06 | -3.14 | -1.67E-06 |

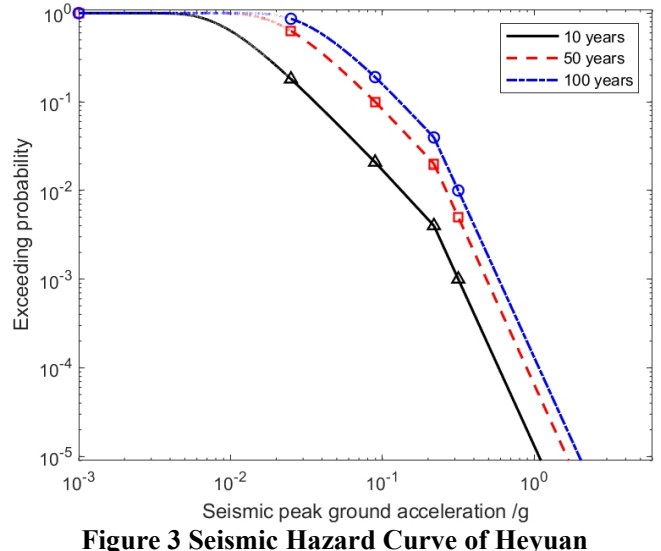
2 **Figure 3 Seismic Hazard Curve of Heyuan**

3          **City**

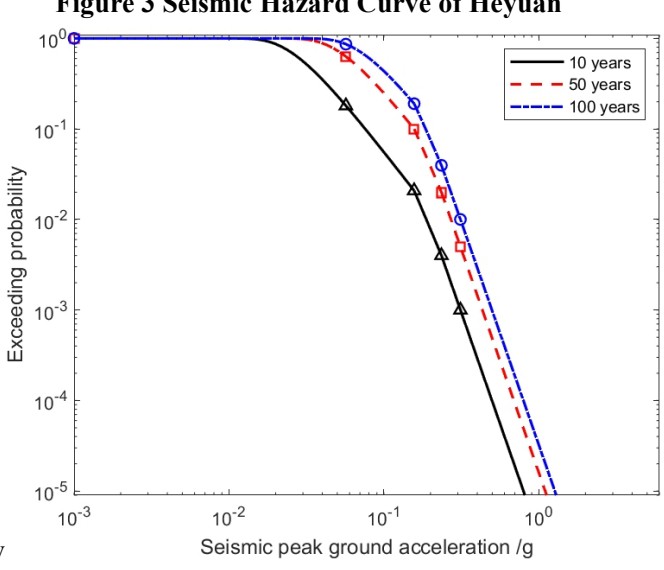

4 **Figure 4 Seismic Hazard Curve of Deyang City**

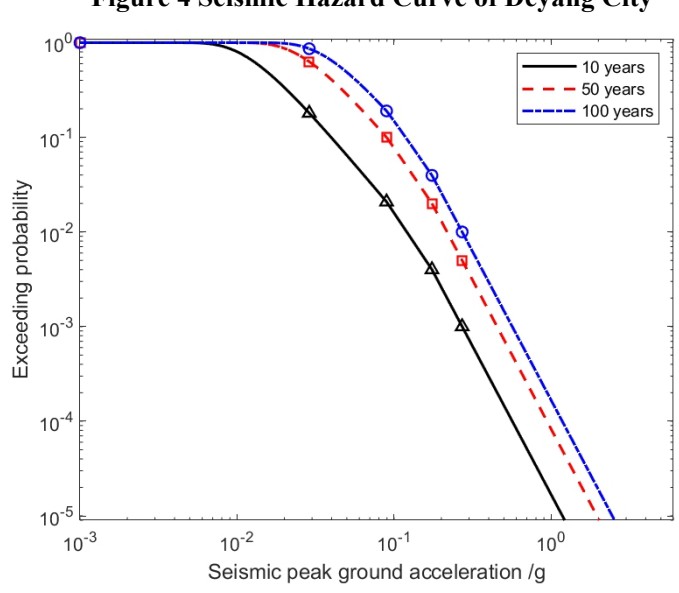

6 **Figure 5 Seismic Hazard Curve of Kelamayi City**

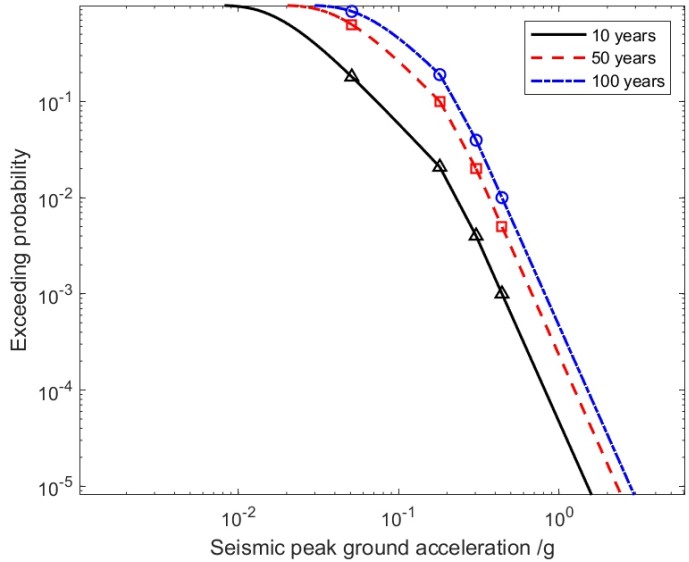

**Figure 6 Seismic Hazard Curve of Mianzhu City**

2.2 Seismic Fragility Analysis of Water Supply Facilities

The main purpose of seismic fragility analysis is to establish the relationship between the probability of water supply system facilities reaching or exceeding different seismic damage levels and ground motion parameters (intensity or peak ground acceleration). The main methods include earthquake damage investigation, theoretical analysis, and experimental analysis.

The water supply system facilities mainly include the water supply pipeline network, water pool, and pump station buildings. In this paper, the water supply pipeline network was divided into five types according to the material: Ductile Cast iron pipe, steel pipe, plastic pipe, reinforced concrete pipe and Cast iron pipe. Each pipe is divided into five different types of seismic capacity zones according to the pipeline's city category, that is, each pipe has a total of five types of fragility curves. When calculating, the corresponding pipe fragility curve must be selected according to the seismic capacity zone of the pipeline's city. The water pool and pump station buildings are divided into two categories based on the seismic capacity zone.

Based on the seismic damage data collected in this article and the "Classification of Seismic Damage Levels in Lifeline Engineering" (GB/T24336-2009 Classification of Earthquake Damage Levels for Lifeline Engineering. (2009).) specification, the seismic damage level of pipelines is determined by the pipeline seismic damage rate. The proportion of pipeline damage levels under the same seismic intensity obtained from seismic damage sample data is the damage ratio in the seismic damage matrix, which then forms the seismic damage matrix for pipelines of various materials.

Based on the seismic damage matrix of the pipeline, the distribution of different damage ratios under different intensities was obtained, and a fitting curve for the damage ratios of different damage levels under different intensities of the water supply pipeline network in 5 levels was established. The fitting results show that its distribution follows the trend of polynomial function distribution.

$$P_i = a_0 d_j^4 + a_1 d_j^3 + a_2 d_j^2 + a_3 d_j + y_0 \qquad (4)$$

Where $P_i$ is the damage ratio of ith level(totally 5 different seismic capacity zones), $d_j$ is jth damage level(Basically intact-1, Slight damage-2, Moderate damage-3, Severe damage-4 and Destroyed-5), $a_0$, $a_1$, $a_2$, $a_3$, $y_0$ are parameters.

We obtained parameters($a_0$, $a_1$, $a_2$, $a_3$, $y_0$ ) through polynomial fitting. Its goodness-of-fit is that the R-square value of polynomials of all pipes is above 0.98.

The seismic risk assessment model for water supply systems proposed in this article involves at least five types of pipeline materials, namely ductile iron pipes, cast iron pipes, steel pipes, PE pipes, and prestressed reinforced concrete pipes. The pipeline fragility curve of each material will be divided into 5 categories according to the seismic capacity zones of cities in Chinese Mainland, because the seismic capacity of Chinese Mainland is divided into 5 zones according to seismic fortification intensity, site claasification and city economic condition in this paper. Due to the fact that the research object of this article is a large-scale water supply network, which is a macro perspective, this article to some extent considers the seismic disaster risk of pipelines caused by fault dislocations. The urban fortification intensity is obtained from the zonation map, which considers factors such as seismic geology of the city, including the impact of faults on urban facilities reflected in seismic fortification. The fragility curves of this article is calculated by fitting the actual seismic damage of pipelines, which includes the damage caused by seismic fault dislocations. As shown in the example of the PE pipe fragility curves in the article, each pipeline material involved in the model in this article will have data similar to the parameters of the PE pipe fragility curve. Due to space limitations, only the fragility curves of PE pipe will be placed in the manuscript.

This article listed the fitting parameters of the damage ratio curve of seismic damage matrix for PE pipelines, as shown in Table 14.

**Table 14 Parameter Values of Damage Ratio Curves for Different Damage Levels of PE Pipe under Different Intensities**

| Seismic capacity level | Parameter | VI | VII | VIII | IX |
|---|---|---|---|---|---|
| Level 1 area | $a_0$ | 4.15 | 0.79 | -0.77 | 0.42 |
| | $a_1$ | -58.03 | -12.85 | 11.99 | -5.50 |
| | $a_2$ | 294.20 | 78.51 | -62.18 | 21.58 |
| | $a_3$ | -638.10 | -215.50 | 111.20 | -27.50 |
| | $y_0$ | 497.50 | 226.00 | -18.20 | 31.00 |
| Level 2 area | $a_0$ | 4.00 | 0.65 | 0.94 | 0.83 |
| | $a_1$ | -56.10 | -10.48 | -8.46 | -11.50 |
| | $a_2$ | 285.30 | 64.70 | 20.31 | 50.67 |
| | $a_3$ | -621.20 | -182.50 | -14.79 | -79.00 |
| | $y_0$ | 487.00 | 200.50 | 38.00 | 54.00 |
| Level 3 area | $a_0$ | 3.83 | 0.04 | 1.88 | 0.63 |
| | $a_1$ | -53.87 | -2.12 | -19.92 | -9.75 |
| | $a_2$ | 274.70 | 23.41 | 67.63 | 47.38 |
| | $a_3$ | -600.60 | -98.03 | -88.58 | -78.25 |

| | | | | | |
|---|---|---|---|---|---|
| | $y_0$ | 474.00 | 143.70 | 71.00 | 50.00 |
| | $a_0$ | 3.50 | -0.17 | 1.67 | -1.04 |
| | $a_1$ | -49.40 | 1.47 | -18.17 | 8.08 |
| Level 4 area | $a_2$ | 253.50 | 1.87 | 62.83 | -14.46 |
| | $a_3$ | -559.60 | -45.37 | -79.33 | 4.42 |
| | $y_0$ | 448.00 | 102.20 | 58.00 | 6.00 |
| | $a_0$ | 3.58 | -0.88 | 2.60 | -2.00 |
| | $a_1$ | -49.43 | 11.12 | -29.79 | 19.07 |
| Level 5 area | $a_2$ | 247.40 | -44.83 | 111.10 | -56.30 |
| | $a_3$ | -533.60 | 47.78 | -153.00 | 66.63 |
| | $y_0$ | 422.00 | 40.80 | 87.00 | -27.00 |

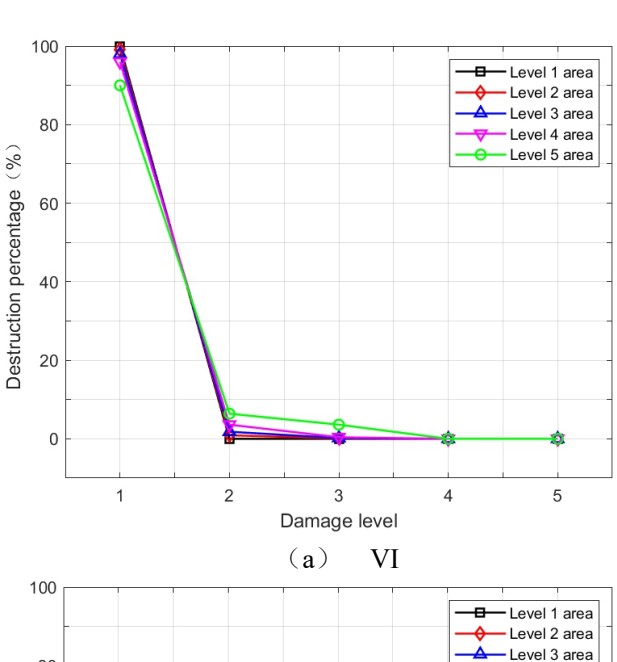

3
(a) VI

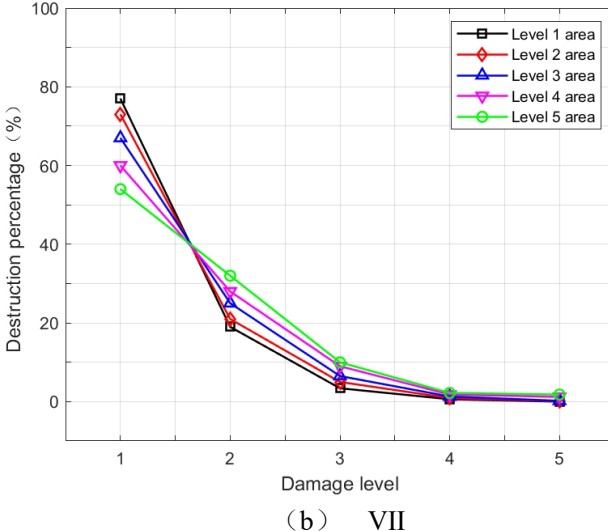

5
(b) VII

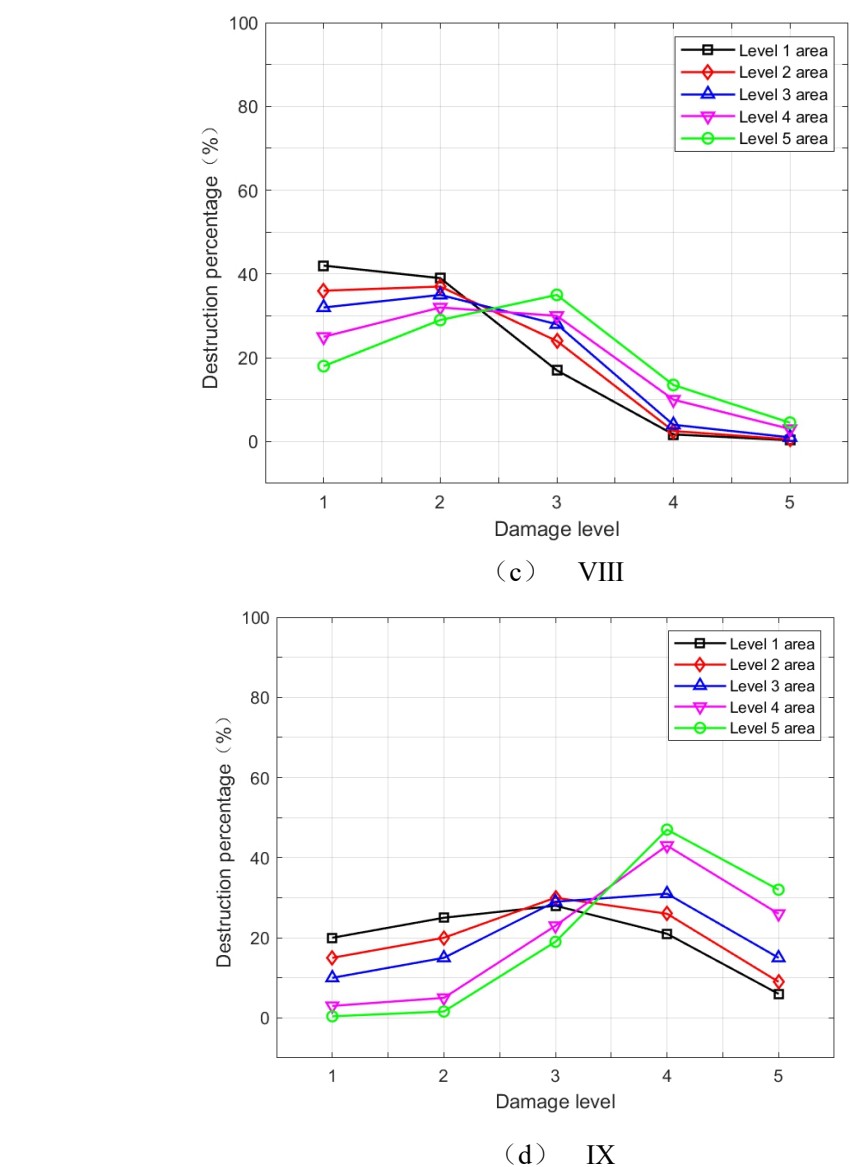

2 (c) VIII

4 (d) IX

**Figure 7 Damage ratio of PE pipe under different seismic intensities(Level 1 area-Level 5 area)**

This article established a seismic fragility function model with the input parameter of seismic peak ground acceleration. The seismic fragility analysis results can generally be represented by the seismic fragility curve or the seismic damage exceeding probability matrix. Therefore, it is necessary to convert the seismic damage matrix based on the peak ground acceleration into the exceeding probability matrix that reaches or exceeds a certain limit state.

In this paper, the logarithmic normal distribution function model (Chen Libo et al., 2012; Chen Bo, 2018) is used as the seismic fragility function $F_m(a)$, $F_m(a)$ is the function of the peak ground acceleration $a$:

$$F_m(a) = \Phi\left[\frac{\ln\left(\dfrac{a}{\theta_m}\right)}{\beta_m}\right] \qquad (5)$$

$a$:Peak ground acceleration,

$m$:Seismic damage level,m=1、2、3、4 and 5 represents damage levels of Basically intact, Slight damage, Moderate damage, Severe damage and Destroyed respectively.

$\Phi$:Standard normal distribution function,

$\theta_m$:The median value of the seismic fragility curve for the m th damage level,

$\beta_m$:Logarithmic standard deviation of seismic fragility curve for the m th damage level.

The probability of being at the m th damage level can be calculated using the following formulas:

$$P_1(\mathrm{D}|a) = 1 - F_2(a) \qquad (6)$$

$$P_m(\mathrm{D}|a) = F_m(a) - F_{(m+1)}(a) \qquad (7)$$

$$P_5(\mathrm{D}|a) = F_5(a) \qquad (8)$$

The two parameters of the seismic fragility function $F_m(a)$ in formula (5) $\theta_m$ and $\beta_m$ is obtained by firstly converting from the pipe seismic damage matrix to the exceeding probability matrix, and then fitting using the least squares method.

This article took the PE pipe as an example and listed the parameters of the seismic fragility function in Table 15. The fragility curve is shown in Figures 8 to 12.

**Table 15 Seismic Fragility Function Parameters of PE Pipe under Different Seismic Capability Levels**

| Seismic capacity level | Parameter | Slight damage | Moderate damage | Severe damage | Destroyed |
|---|---|---|---|---|---|
| 1 | $\theta$ | 0.2466 | 0.4724 | 0.8187 | 1.3791 |
|   | $\beta$ | 0.8000 | 0.8000 | 0.6500 | 0.5952 |
| 2 | $\theta$ | 0.2255 | 0.4066 | 0.7047 | 1.1724 |
|   | $\beta$ | 0.7500 | 0.7500 | 0.6667 | 0.6427 |
| 3 | $\theta$ | 0.1993 | 0.3234 | 0.5488 | 0.8607 |
|   | $\beta$ | 0.6333 | 0.7000 | 0.6800 | 0.5302 |
| 4 | $\theta$ | 0.1597 | 0.2594 | 0.4066 | 0.7469 |
|   | $\beta$ | 0.7446 | 0.6574 | 0.7000 | 0.6539 |
| 5 | $\theta$ | 0.1319 | 0.2466 | 0.3679 | 0.6703 |
|   | $\beta$ | 0.5391 | 0.5600 | 0.5800 | 0.7000 |

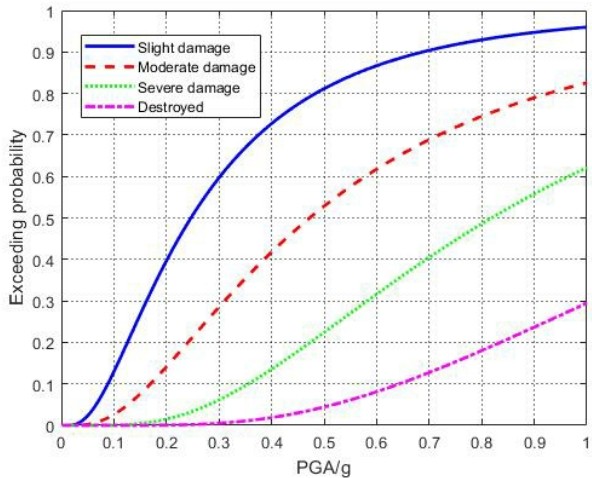

Figure 8 Fragility Curve of PE Pipe in Seismic Capacity Level 1 Area

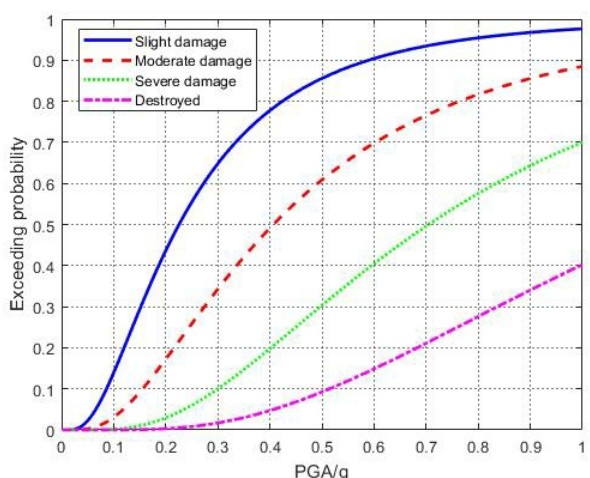

Figure 9 Fragility Curve of PE Pipe in Seismic Capacity Level 2 Area

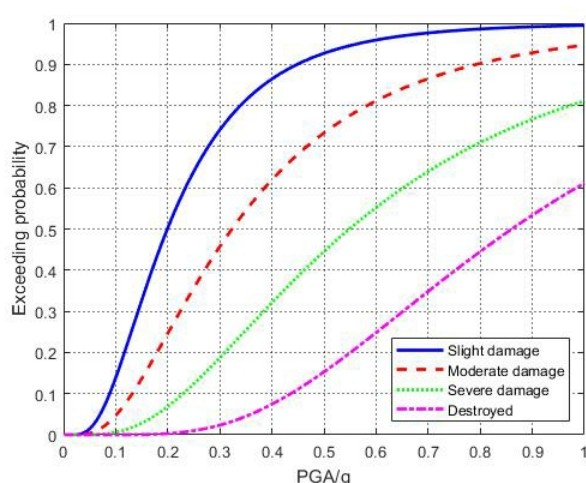

Figure 10 Fragility Curve of PE Pipe in Seismic Capacity Level 3

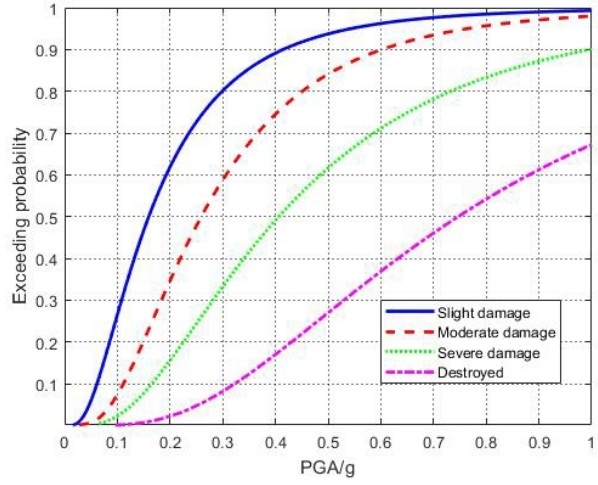

**Area**

**Figure 11 Fragility Curve of PE Pipe in Seismic Capacity Level 4**

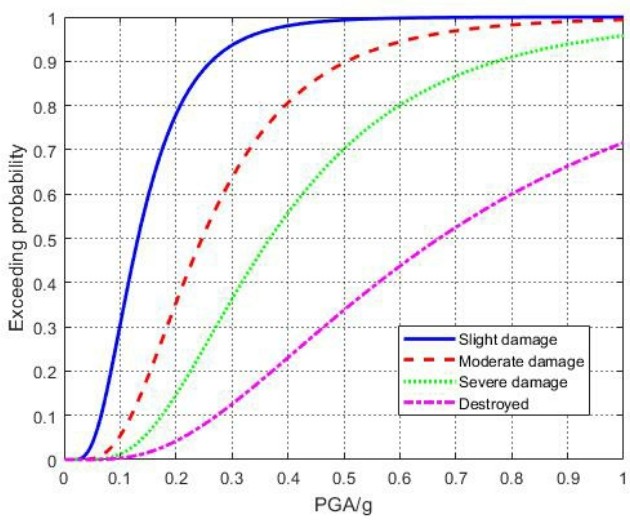

**Area**

**Figure 12 Fragility Curve of PE Pipe in Seismic Capacity Level 5 Area**

2.3 Water supply system exposure

Before assessing the seismic risk in the water supply system, it is necessary to

know the exposure of the water supply system. The total fixed assets of the water

supply system, as the quantitative characteristics of the expected loss caused by the

possible earthquake disaster in the region, can represent its exposure. Using the

"Water Supply Yearbook" to collect the total fixed assets of the regional water supply

system, it is necessary to know the proportion of water supply network, pool, and

pump station building assets in the total fixed assets. Based on literature statistics and

analysis, this article determined that in the water supply system, pipeline assets

account for 70%, pool assets account for 22% (with clean water pools and water

treatment pools each accounting for 50% of pool assets), and pump station buildings

account for 8%. (Fan Wenting, 2020; Nong Weiwen, 2006; China Water Supply

Association, 2009)

2.4 Comparison with actual earthquake damage losses

When the water supply system encounters a seismic peak ground acceleration of $a$, the loss is (Yin Zhiqian, 2004):

$$L(a) = \sum_s \sum_m (W_s r_{ms}) P_{ms}(\mathrm{D}|a) \qquad (9)$$

$L(a)$:The loss of the water supply system when encountering a seismic peak ground acceleration of $a$.

$W_s$:Total replacement cost of Class S water supply system facilities.

$r_{ms}$:The loss ratio of Class S water supply system facilities in the M damage level,

$P_{ms}(\mathrm{D}|a)$:The probability of Class S water supply system facilities experiencing M damage level when peak ground acceleration is $a$.

According to the seismic hazard curve of Deyang City, combined with the seismic fragility of various facilities of the water supply system and the distribution of various facilities assets of the water supply system, the 50-year exceeding probabilities of 63%, 10% and 2% were respectively predicted, which corresponded to the earthquake disaster losses of the water supply system in Deyang City when the intensity was VI, VII and VIII. The actual earthquake losses and predicted losses are shown in Table 16.

**Table 16 Earthquake disaster Loss Prediction of Deyang Water Supply System**

| Intensity | Actual losses (10,000 yuan) | Predicted losses (10,000 yuan) | 50-year exceeding probability(%) |
|---|---|---|---|
| VI | | 613 | 63 |
| VII | 3500 | 3394 | 10 |
| VIII | | 5634 | 2 |

The probability of occurrence of intensity VI and VII in Deyang City is 39.24% and 24.63%, respectively, which are one to two orders of magnitude higher than the probabilities of occurrence of other intensities. This indicates that the seismic intensity threat in Deyang City in the next 50 years mainly comes from intensity VI and VII. Although the exceeding probability of degree VI is 63%, which belongs to the level of frequent seismic motion, the predicted loss of degree VI is less than that of degree VII by one order of magnitude, and the destructive effect is relatively small. Although the predicted loss of degree VIII is greater than that of degree VII, the exceeding probability of degree VIII is only 2%, which belongs to the level of rare seismic motion. Therefore, the seismic risk faced by Deyang City is mainly the earthquake loss caused by intensity VII. The predicted loss of intensity VII in Deyang City is 33.94 million yuan, which is more consistent with the actual loss of 35 million yuan caused by Wenchuan earthquake. This confirms the reliability of the seismic fragility function proposed in this article.

# 3 Seismic risk distribution based on loss (rate) expectation in water supply systems

Using the seismic hazard analysis method, the seismic fragility model of water supply system and the distribution of fixed assets introduced in Part 2, the loss expectation and loss rate expectation of earthquake disaster in a certain area at different time scales were calculated. In the scale of future t years, the full probability of the class s water supply system facilities experiencing m damage level is:

$$PDf_{ms} = \int P_{ms}\left(D|a\right)f_t\left(a\right)da \qquad (10)$$

$PDf_{ms}$: Full probability of the class s water supply system facilities experiencing m damage level in future t years,

$P_{ms}\left(D|a\right)$: The probability of Class S water supply system facilities experiencing M damage level when peak ground acceleration is $a$.

$f_t\left(a\right)$: Probability density function of peak ground acceleration in future t-year scale.

At the scale of t years in the future, the loss expectation of water supply system facilities caused by the peak ground acceleration of various intensities that may occur in the local area is expressed as the sum of the product of direct loss when the s-class water supply system facilities experience m damage level and the full probability. The calculation model is:

$$E\left[L_t\right] = \sum_s \sum_m (W_s r_{ms})PDf_{ms} \qquad (11)$$

$E[L_t]$ is water supply systems loss expectation in the future t years.

$W_s$ is total replacement cost of s-class water supply system facilities (s-class total fixed assets).

$r_{ms}$ is the loss ratio of s-class water supply system facilities in the m damage level.

For example, let's assume that the probability of a specific damage level occurring at the peak ground acceleration $a$ of Class S water supply facilities is $P_{ms}\left(D|a\right)$, and this specific damage level is assumed to be m (a total of five damage levels, with a sum of 1 at the same peak acceleration). The economic loss when a specific damage level m occurs is the product of the total asset cost $W_s$ and the loss ratio $r_{ms}$. Due to the fact that under a specific peak ground acceleration $a$, the probability of m damage level occurring is not 1, but $P_{ms}\left(D|a\right)$. Therefore, under a peak acceleration $a$, the loss of a water supply facility with m damage level occurring is $W_s r_{ms} P_{ms}\left(D|a\right)$ ( equation 1). According to seismic hazard analysis, the

exceeding probability of peak ground acceleration $a$ at a certain time scale can be converted into the cumulative distribution probability of peak acceleration $a$. The probability density function $f_t(a)$ of peak acceleration $a$ can be obtained by calculating the first derivative of the cumulative distribution probability function. Due to the fact that the probability density function is a continuous function rather than a step function, the probability of the occurrence of peak acceleration $a$ can be considered as $f_t(a)da$. For a complete seismic risk assessment, the possibility of the earthquake itself should be considered. The loss caused by the damage level m of s type water supply facilities should be multiplied by the probability of the occurrence of peak acceleration a based on equation 1, that is, $W_s r_{ms} P_{ms}(D|a) f_t(a) da$ ( equation 2). Due to the uncertainty of earthquake occurrence, each peak acceleration $a$ has a certain probability of occurrence. Therefore, equation 2 is summed in the direction of acceleration a, $\int W_s r_{ms} P_{ms}(D|a) f_t(a) da = (W_s r_{ms}) \int P_{ms}(D|a) f_t(a) da$. The total expected loss caused by various damage levels and types of water supply facilities is:

$$E[L_t] = \sum_s \sum_m (W_s r_{ms}) PDf_{ms}.$$

In the scale of t years in the future, the loss expectation of water supply system facilities caused by peak ground accelerations of various intensities that may occur in the local area divided by the total cost of resetting the water supply system facilities in the local area is loss rate expectation:

$$E[R_t] = \frac{E[L_t]}{\sum_s W_s} \qquad (12)$$

China's capital circle, southern Liaoning, north-south seismic belt, northwestern Xinjiang, Yangtze River Delta and Pearl River Delta regions, and most provincial capital cities have high seismic loss expectations. The high level of seismicity and the high risk of seismic hazard are the main reasons for Xinjiang and the north-south seismic belt; The eastern region is due to its developed economy, high level of urbanization, abundant water supply system facilities, and high exposure of disaster bearing bodies; The capital circle and southern Liaoning region are the results of the combination of seismic hazard and exposure of disaster bearing bodies. The top 10 cities in descending order of loss expectation are Beijing, Kunming, Tianjin, Shanghai, Guangzhou, Guyuan, Shenyang, Chengdu, Ningbo, and Xi'an. Among them, mega cities may not necessarily be in seismic hazard areas, such as Shanghai and Guangzhou, mainly due to the large stock of water supply networks in mega cities and the high asset value affecting loss expectations. Cities with high loss rate expectation are generally located in seismic hazard areas or have high seismic fragilities, not only affected by the large stock of water supply networks and high assets; Moreover, the

loss rate expectation is expected to have exponential characteristics, which can be used as a regional seismic risk index to compare the seismic risk between cities.

Considering the difference between the seismic loss rate expectation of the water supply system or a certain facility in different time scales due to the seismic hazard probabilities, the 10-year scale and 100-year scale standards adopt the 50-year scale seismic loss rate expectation index classification standard.

For the 50-year scale, considering that the seismic loss rate expectation of the water supply system is independent, when determining the classification standard of the seismic loss rate expectation index of the water supply system, this paper divided the classification standard of the seismic loss rate expectation index of the water supply system according to the principle that the number of cities in all categories accounts for basically the same proportion.

The seismic loss rate expectation of the water supply system can be used as the regional seismic risk index to compare the seismic risk between cities, so as to carry out the seismic risk assessment for the regional water supply systems.

**Table 17 Grade classification standard of seismic loss rate expectation index of regional water supply systems**

| Classification of loss rate expectation | A | B | C | D | E |
|---|---|---|---|---|---|
| Loss rate expectation index | [0.085-1.0) | [0.030-0.085) | [0.018-0.030) | [0.0075-0.018) | (0-0.0075) |
| Risk level | Very high | High | Medium | Low | Very Low |
| Symbol color | Red | Orange | Yellow | Blue | Green |

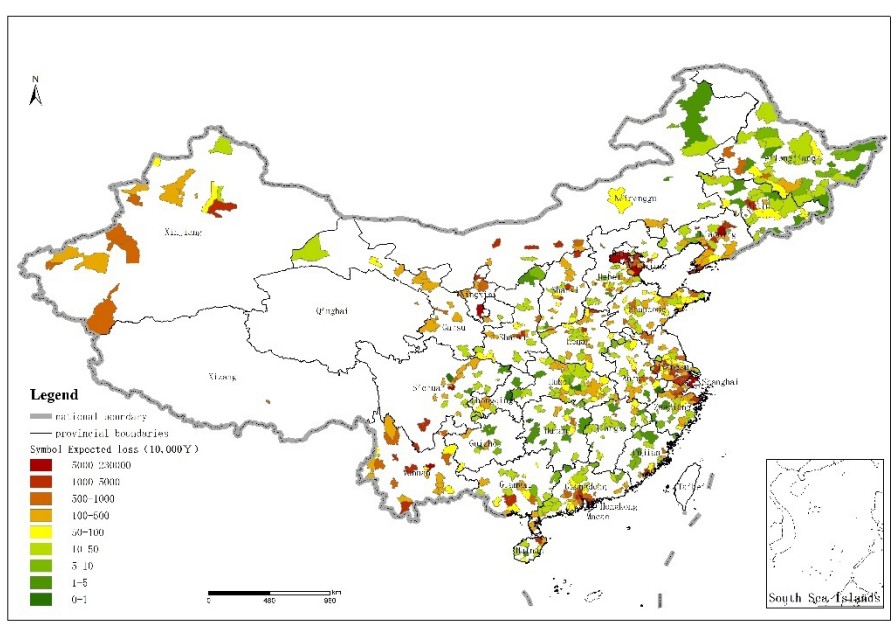

2      **Figure 13 Distribution Map of 10-year seismic loss expectation of Water Supply Systems in**
3                          **720 cities in Chinese Mainland**

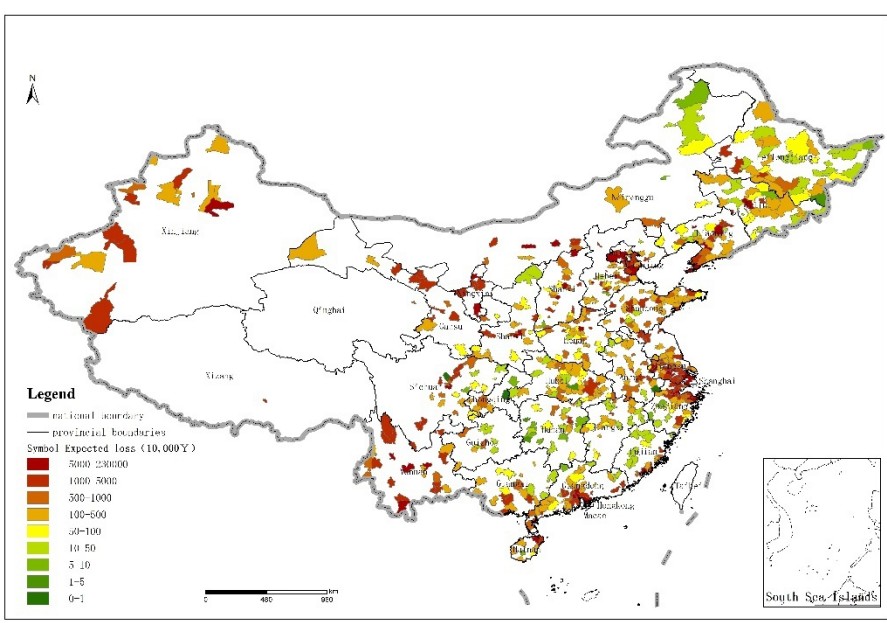

5      **Figure 14 Distribution Map of 50-year seismic loss expectation of Water Supply Systems in**
6                          **720 cities in Chinese Mainland**

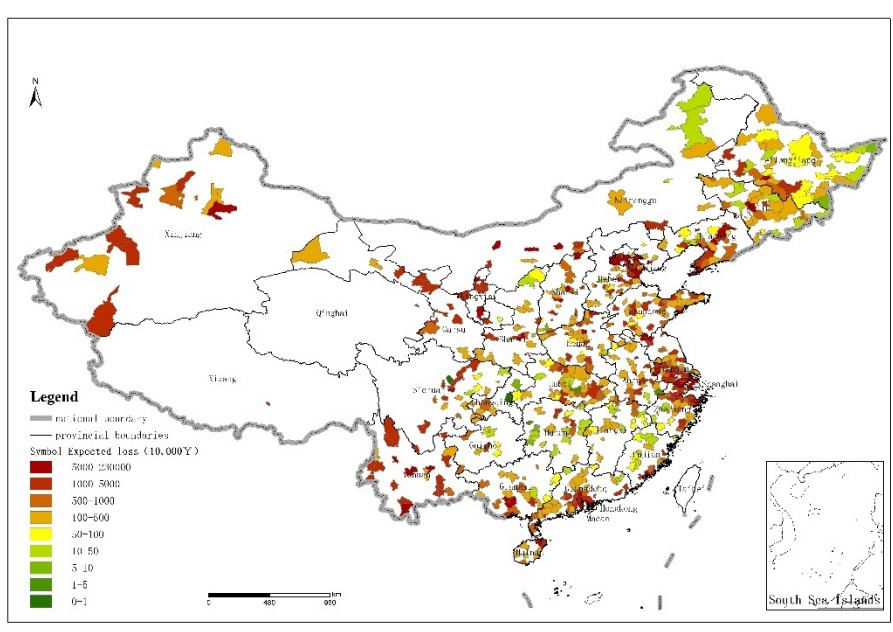

2    **Figure 15 Distribution Map of 100-year seismic loss expectation of Water Supply Systems in**
3                          **720 cities in Chinese Mainland**

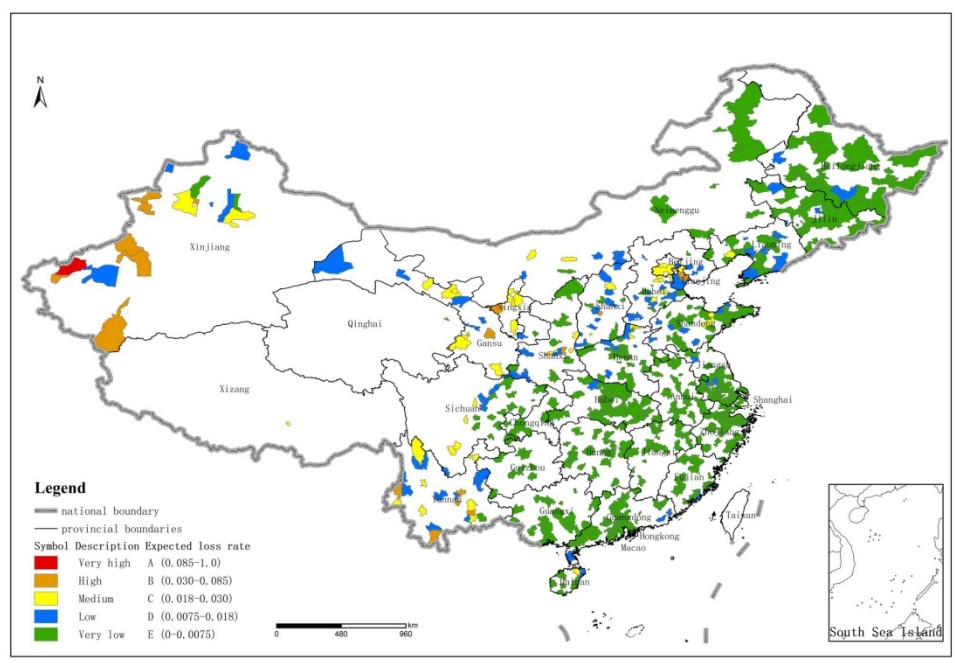

5    **Figure 16 Distribution Map of 10-year seismic loss rate expectation of Water Supply Systems**
6                          **in 720 cities in Chinese Mainland**

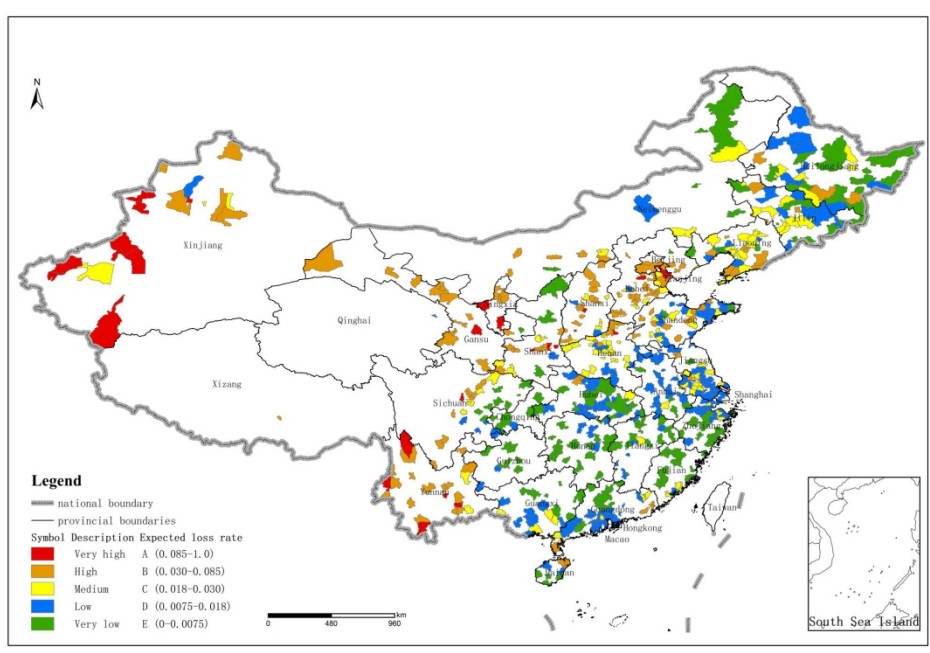

2 **Figure 17 Distribution Map of 50-year seismic loss rate expectation of Water Supply Systems**

3 **in 720 cities in Chinese Mainland**

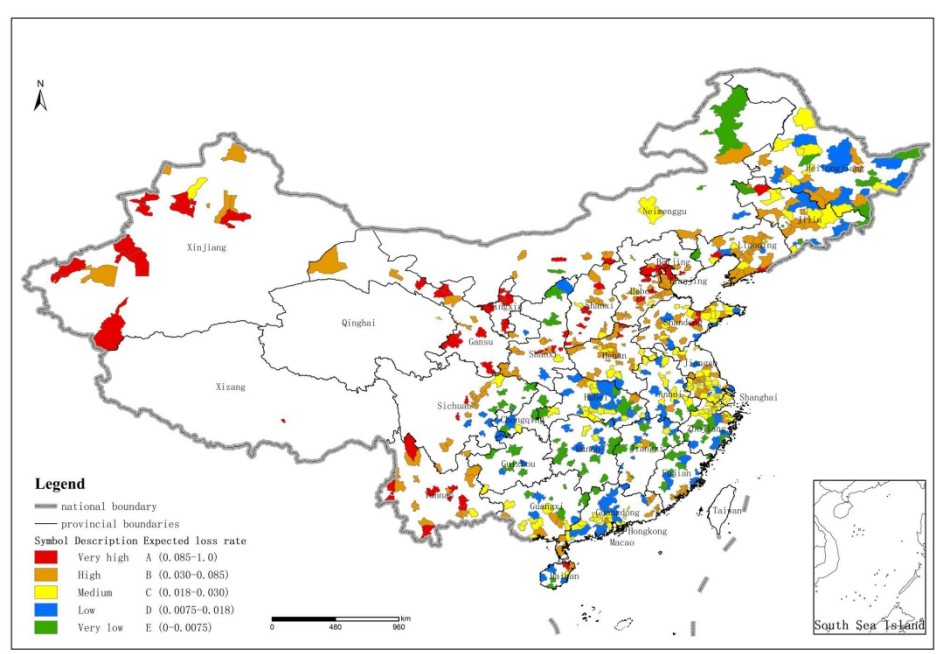

5 **Figure 18 Distribution Map of 100-year seismic loss rate expectation of Water Supply**

6 **Systems in 720 cities in Chinese Mainland**

The seismic disaster risk assessment model for water supply systems proposed in this article is an assessment of the uncertainty of the occurrence of seismic disasters in water supply systems, and model validation should adopt a qualitative approach. This model covers the levels of ground motion at the probability levels of frequent, basic, rare, and extremely rare occurrences. Therefore, taking the Wenchuan 8.0 earthquake that occurred on May 12, 2008 as an example, this article used the model to calculate the seismic loss rate expectation and risk levels of water supply systems in 5 cities in Sichuan Province and 1 city in Shaanxi Province before the earthquake, as shown in Table 11. For the convenience of verifying the rationality of the model, Table 18 listed the leakage rates of the water supply systems before and after the earthquake, the basic seismic ground motion (pre-earthquake fortification intensity), the on-site investigation seismic intensity, and the evaluated earthquake damage degree (Institute of Engineering Mechanics, China Earthquake Administration, 2009). It can be seen from Table 4-3 that the post-earthquake on-site investigation intensities of the listed cities are to varying degrees greater than the pre-earthquake fortification intensities. Among them, the post-earthquake intensity of Mianzhu and Dujiangyan exceeded the pre-earthquake intensity by 2 degrees, and the pre-earthquake predicted seismic risk level are the highest (Grade A). The post-earthquake intensity of Jiangyou, Mianyang, Guangyuan and Ningqiang exceeded the pre-earthquake intensity by 1 degree. The pre-earthquake predicted seismic risk levels are Grade B and Grade C, although it is lower than that of the first two cities, However, they are still at high and medium risk levels, respectively. In addition, cities with a predicted seismic risk level A of water supply systems before the earthquake correspond to the earthquake intensity of "IX" and the earthquake damage level of "destruction" surveyed on site after the earthquake; Cities with seismic disaster risk level B correspond to the seismic intensity of "VIII" and seismic damage level of "severe damage" in the post-earthquake on-site investigation; Cities with a seismic disaster risk level of C correspond to the seismic intensity of "VII" and the seismic damage level of "moderate damage" or "slight damage" according to the on-site investigation after the earthquake. The validation results indicate that the proposed water supply systems risk model can accurately predict the level of seismic risk faced by urban water supply systems in China.

**Table 18 Comparison between the Wenchuan 8.0 earthquake damage and predicted seismic risk levels**

| City | Pre-earthquake leakage rate(%) | Post-earthquake leakage rate(%) | Basic seismic ground motion (fortification intensity) | On site investigation seismic intensity | Seismic damage level | Pre-earthquake loss rate expectation index | Pre-earthquake risk level description |
|---|---|---|---|---|---|---|---|
| Mianzhu | 17 | 85 | VII | IX | Destroyed | 0.111 | Very high(A) |
| Dujiangyan | 27 | 60 | VII | IX | Destroyed | 0.087 | Very high(A) |
| Jiangyou | 26 | 50 | VII | VIII | Severe damage | 0.032 | High(B) |
| Mianyang | 12 | 17 | VI | VII | Moderate damage | 0.019 | Medium(C) |
| Guangyuan | 21 | 24 | VI | VII | Moderate damage | 0.018 | Medium(C) |
| Ningqiang | 20 | 25 | VI | VII | Slight damage | 0.018 | Medium(C) |

In order to illustrate the rationality of the classification of the seismic loss rate expectation index of the water supply systems in Chinese Mainland, the research results of China's seismic hazard and key monitoring and defense areas from 2006 to 2020 (Wang Xiaoqing, 2006) are introduced. The key hazard areas and seismic damage prediction results are the main basis for determining the key monitoring and defense areas in China from 2006 to 2020. The key monitoring and defense areas are determined based on comprehensive consideration of the earthquake situation, disaster situation, and social development. Among them, the prediction results of earthquake life and economic losses are the most important basis for determining the key monitoring and defense areas.

As shown in Figure 19, the country is divided into four seismic hazard areas and key monitoring and defense areas (areas surrounded by the blue line in the figure). 1 represents North China (Beijing, Tianjin, Hebei, Shanxi, and southern Liaoning), 2 represents the north-south belt region (Gansu, Qinghai, Ningxia, Shaanxi, Sichuan, Yunnan), 3 represents the northwest region of Xinjiang, and 4 represents the southeast coastal region (Fujian Guangdong border area, Taiwan Strait, Haikou City, Hainan

Province).

The loss rate expectation index and seismic risk levels of the water supply systems are relatively high in the four seismic hazard and key monitoring and defense areas mentioned above.This result is consistent with the research results of seismic hazard and key monitoring and defense areas in China from 2006 to 2020. As shown in Figure 19. Because the above areas are located in the seismic zone, seismicity is frequent and the seismic hazard is high.

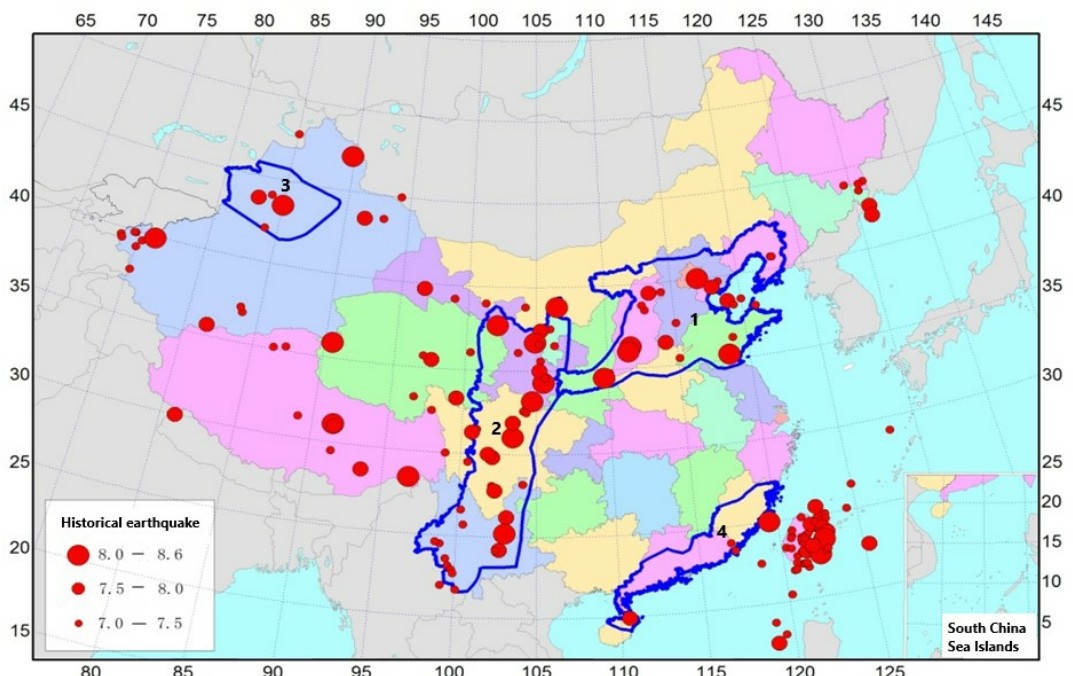

**Figure 19 Seismic hazard and key monitoring and defense areas in China from 2006 to 2020**

## 4 Discussion

In terms of the research on the resilience of post-earthquake water supply networks, this article introduced the concept of recovery difficulty to evaluate the resilience of water supply networks after earthquakes. Recovery difficulty index could be calculated as follows:

$$R_d = \frac{Q_{La} - Q_{Lb}}{Q_{Lb}}$$

(13)

$Q_{La}$—Post-earthquake leakage rate(%);

$Q_{Lb}$—Pre-earthquake leakage rate(%).

This indicator has low requirements for the completeness of statistical data in practical operation, therefore it has practical engineering value. The changes in the leakage rate of water supply pipelines before and after an earthquake can reflect the damage situation of the pipeline network. The greater the leakage rate of the pipeline network after an earthquake, the more severe the damage to the pipeline network, and the greater the difficulty of recovery.

It can be seen from Table 19 that Mianzhu and Dujiangyan, where the seismic intensity was 9, the networks fortification intensity was 7, and the water supply networks damage level was destroyed, were the most difficult to recover. Secondly, Jiangyou, with an seismic intensity of 8, network fortification intensity of 7, and water supply network damage level of severe damage. The seismic intensity of Mianyang, Guangyuan, and Ningqiang was 7, and the seismic fortification intensity of the networks was 6. The difficulty of recovering the water supply networks after the earthquake was relatively low.

**Table 19 Damage and Recovery Difficulty Index of Water Supply Networks in Wenchuan Earthquake**

| City | Pre-earth quake leakage rate(%) | Post-earth quake leakage rate(%) | Basic seismic ground motion (fortification intensity) | On site investigation seismic intensity | Seismic damage level | Recovery difficulty index |
|---|---|---|---|---|---|---|
| Mianzhu | 17 | 85 | VII | IX | Destroyed | 4.00 |
| Dujiangyan | 27 | 60 | VII | IX | Destroyed | 1.22 |
| Jiangyou | 26 | 50 | VII | VIII | Severe damage | 0.92 |
| Mianyang | 12 | 17 | VI | VII | Moderate damage | 0.42 |
| Guangyuan | 21 | 24 | VI | VII | Moderate damage | 0.14 |
| Ningqiang | 20 | 25 | VI | VII | Slight damage | 0.25 |

## 5 Conclusion

This paper proposed an assessment model based on loss expectation and loss rate expectation for seismic risk assessment of water supply system in Chinese Mainland. This model solves the different needs of government departments for the risk level of seismic risk in the water supply system, and provides technical support for the risk

zonation and risk mapping of earthquake disaster in the water supply system. The specific conclusions obtained through this study are as follows:

1) Based on multi-source basic data such as urban industry yearbook, seismic zonation, seismic code, population GDP and historical earthquake damage data, a basic database for seismic risk assessment of 720 urban water supply systems in Chinese Mainland was established. The probability density functions of peak ground acceleration were calculated by using the seismic hazard analysis method, and the parameters of the seismic risk curves of 720 cities were calculated. The seismic damage matrix of pipelines and facilities is obtained based on the actual seismic damage through statistical calculation, and the seismic fragility curves of various facilities in the water supply system were given based on the logarithmic normal distribution model.

2) The risk index of earthquake disaster is the result of the joint action of earthquake occurrence probability, vulnerability and exposure. The seismic loss rate expectation index is used as the seismic risk assessment index to evaluate the water supply systems. The grading evaluation criteria of risk index (A-E) were established, and the distribution maps of seismic loss expectation and the classification maps of loss rate expectation index of 720 urban water supply systems in Chinese Mainland in medium and long-term were given.

3) According to the conclusion that the region where the cities with risk levels A and B are located is more consistent with the research results of China's seismic hazard and key monitoring and defense areas from 2006 to 2020, it shows that the seismic risk assessment of regional water supply systems is highly correlated with the medium and long-term earthquake prediction results, which is suitable for the medium and long-term risk assessment, and verifies the rationality and applicability of the model proposed in this paper. In particular, we should strengthen the prevention and control of seismic risk in key cities in North China, Northwest China, Southwest China and South Northeast China, and improve the seismic capacity of water supply systems and facilities in these key risk cities.

## Data availability

The datasets used in the study were derived from the following resources available in the public domain: Communiqué of the National Bureau of Statistics of the People's Republic of China on Major Figures of the 2010 Population Census, Statistical Yearbook of Urban Water Supply (2009-2018), GB50011-2010Code for seismic design of buildings (2010), Summary report on scientific investigation of earthquake damage in Wenchuan earthquake, GB18306-2015Seismic ground motion parameters zonation map of China. (2015). Site category data was calculated through BP neural network method. Seismic hazard control points were calculated using CPSHA method. Both site category and seismic hazard control points data are classified and could not be available in the public domain.

## Author contributions

Tianyang Yu initiated the research. Tianyang Yu and Banghua Lu gathered the data. Tianyang Yu analyzed the data and plotted the maps and graphs. Tianyang Yu wrote the manuscript draft. Hui Jiang and Zhi Liu reviewed the manuscript.

## Competing interests

The contact author has declared that none of the authors have any competing interests.

## Acknowledgement

This research is supported by "National Natural Science Foundation of China (U1901602) ", "Southern Marine Science and Engineering Guangdong Laboratory (Zhuhai) ( No.  SML2023SP206) " and "Seismological Research Foundation for Youths of Guangdong Earthquake Agency(GDDZY202308)".

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
