# Peer review of "Study on Seismic Risk Assessment Model of Water Supply"

_Natural Hazards and Earth System Sciences, 2023_

## Author Comment (AC1)

Also, I suggest the authors add a graphical presentation of the research at the beginning of the paper showing inputs, methods and outputs with respective interrelations. That would help readers to follow the elaborate calculations.

[Figure]

**Figure 1 Flow Chart of Seismic Risk Assessment for Water Supply Systems**

Adding a Discussion chapter can substantially improve the paper, especially in addressing the earthquake resilience of Chinese cities regarding water supply.

[revised manuscript text omitted]

**1. **Technical corrections**

There is a need for technical corrections. The page numeration should be continuous. There seems to be missing text at the end of page 4. The authors should carefully check the paper for unnecessary long sentences and word choices. A more detailed technical review will follow at a later stage.

The initial letter of word 'parameters' was incorrectly capitalized. So it seemed to be two sentences. In fact, "Where $a$ is peak ground acceleration, $t$ is Time (year), $k_b$ and $k_H$ is parameters of seismic hazard curve.' is one sentence. I will carefully check the paper.

---

## Author Comment (AC2)

The question you raised is very good. Please allow me to provide the following explanation:

1. The research object of this paper is the large regional water supply network, involving 720 cities in Chinese Mainland. Treating each city as a point is a macro scale study. The analysis of the impact of fault factors on pipeline seismic risk that you mentioned requires knowledge of the specific location of each pipeline and the specific location of faults in a city. It is a study that has detailed basic data on pipelines and faults for a specific city. I once worked on an engineering project where the basic data of 35555 main water supply pipelines in Guangzhou, China included their specific locations, as well as knowing the location of faults. Then, I considered the impact of fault dislocations on pipeline seismic damage and risk. But that is not the research object of this article. The area involved in this study is very large, so it is difficult to collect data from Guangzhou city widely across 720 cities. Therefore, the main approach adopted in this article is to study the composition and proportion of pipeline materials and diameters for a city. According to each type of pipeline, the seismic fragility curve and seismic hazard curve are coupled to calculate the seismic risk.

2.The pipeline seismic fragility curve used in the risk assessment in the article takes into account the actual seismic damage of the pipeline during an earthquake, including the situation of pipeline seismic damage caused by fault action. Therefore, this article considers the fragility curves of earthquake damage experience to some extent, taking into account the impact of pipeline seismic damage caused by fault action.

3.The current seismic design code for pipelines in China was implemented in 2003. It clearly stipulates the safe distance that various pipelines need to avoid when encountering faults. That is to say, most pipelines nowadays have already taken into account and dealt with the avoidance of active faults during construction.

4. The issue of chain economic losses caused by pipeline earthquake damage that you mentioned also requires detailed data on each pipeline in a city, accurate to the function and location of each pipeline. This topic is suitable for the study of detailed and accurate pipeline basic data in a small area or city. I have done similar research in my doctoral dissertation, taking Karamay City in China as an example, to study the seismic risk transmission of water supply network. At that time, there were relatively detailed basic data on the city's pipelines. Due to limitations in research topic and length, this article did not conduct research on this issue. If possible, I can write a specialized article on this topic to study post-earthquake chain economic losses.